# Therapeutic Effects of IL-1RA against Acute Bacterial Infections, including Antibiotic-Resistant Strains

**DOI:** 10.3390/pathogens13010042

**Published:** 2023-12-31

**Authors:** Ines Ambite, Thi Hien Tran, Daniel S. C. Butler, Michele Cavalera, Murphy Lam Yim Wan, Shahram Ahmadi, Catharina Svanborg

**Affiliations:** Division of Microbiology, Immunology and Glycobiology, Department of Laboratory Medicine, Faculty of Medicine, Lund University, 221 84 Lund, Sweden; ines.ambite@med.lu.se (I.A.); hien.tran@med.lu.se (T.H.T.); daniel.butler87@gmail.com (D.S.C.B.); michele.cavalera@med.lu.se (M.C.); murphy.wan@med.lu.se (M.L.Y.W.); shahram.ahmadi@med.lu.se (S.A.)

**Keywords:** infection, immunotherapy, antibiotic resistance, urinary tract infection, uropathogenic *Escherichia coli*, IL-1 receptor antagonist, IL-1, substance P, acute cystitis, acute pyelonephritis

## Abstract

Innate immunity is essential for the anti-microbial defense, but excessive immune activation may cause severe disease. In this study, immunotherapy was shown to prevent excessive innate immune activation and restore the anti-bacterial defense. *E. coli*-infected *Asc*^−/−^ mice develop severe acute cystitis, defined by IL-1 hyper-activation, high bacterial counts, and extensive tissue pathology. Here, the interleukin-1 receptor antagonist (IL-1RA), which inhibits IL-1 hyper-activation in acute cystitis, was identified as a more potent inhibitor of inflammation and NK1R- and substance P-dependent pain than cefotaxime. Furthermore, IL-1RA treatment inhibited the excessive innate immune activation in the kidneys of infected *Irf3*^−/−^ mice and restored tissue integrity. Unexpectedly, IL-1RA also accelerated bacterial clearance from infected bladders and kidneys, including antibiotic-resistant *E. coli*, where cefotaxime treatment was inefficient. The results suggest that by targeting the IL-1 response, control of the innate immune response to infection may be regained, with highly favorable treatment outcomes, including infections caused by antibiotic-resistant strains.

## 1. Introduction

Bacterial infections continue to challenge human health as a major cause of morbidity and are increasingly difficult to treat. The World Health Organization (WHO) has declared antibiotic resistance as one of the “biggest threats to global health, food security, and development today” [1], highlighting the need for novel solutions. Experimental studies have identified immunotherapy as an interesting new treatment approach, with significant potential [2,3,4], but there is a need to further understand the suitability and specificity of different molecular solutions and to compare immunotherapy to antibiotics, especially as an approach to treating infections caused by antibiotic-resistant bacterial strains.

The potential of immunotherapy has been extensively studied in models of urinary tract infections (UTIs) [5]. Genetic screens have identified single genes that determine the susceptibility to acute pyelonephritis or acute cystitis [6]. The closely related transcription factors interferon regulatory factor (IRF)-3 and IRF-7 form heterodimers that regulate the activation of type I interferons (IFNs) and IFN-dependent genes, especially during viral infections [7,8]. The IRFs have subsequently been shown to have a broader effect on innate immune responses, also including bacteria [9]. *Irf3* deletions create a dramatic renal disease phenotype in infected mice due to the loss of the IRF3-dependent defense, including IFN-β, and the overexpression of *Irf7* and IRF7-dependent gene networks [3,8]. The importance of IRF-7 as a disease regulator is further emphasized by the complete lack of disease response in infected *Irf7*^−/−^ mice. Deletions of the inflammasome constituents *Asc* (apoptosis-associated speck-like protein containing a CARD or *Pycard*) or *Nlrp3* (NLR family pyrin domain containing 3) create an equally dramatic acute cystitis phenotype, caused by excessive activation of interleukin-1 (IL-1) and the pain sensors neurokinin 1 receptor (NK1R) and substance P (SP), characterized by mucosal hyper-inflammation and pain [2,4].

The feasibility of targeting the disease-associated genes was supported by the use of *Irf7* siRNA therapy, which inhibited the disease response in the kidneys of susceptible *Irf3*^−/−^ mice [3], and of a recombinant IL-1 receptor antagonist (IL-1RA) [2], which inhibited acute cystitis in *Asc*^−/−^ mice by blocking the hetero-dimerization of the IL-1 receptor and its accessory proteins (IL-1R1 and IL-1RAcP) [10]. This study focused on the accelerated bacterial clearance, which previously was shown to accompany immunotherapy with *Irf7* siRNA, IL-1RA, and NlpD treatment [2,3,11], extending the analysis to include antibiotic-resistant *Escherichia coli* strains. The results demonstrate strong protective effects of IL-1RA, including effects on infection with four antibiotic-resistant *E. coli* strains that produce extended-spectrum β-lactamase (ESBL). The ESBL-producing clinical strains did not respond to the antibiotic cefotaxime, which was used as a comparator in these studies. The results further demonstrate the potent effects of IL-1RA treatment against acute pyelonephritis in susceptible *Irf3*^−/−^ mice. The findings are conceptually important because immunotherapy showed strong combined effects against infection and inflammation, preventing disease in kidneys and bladders.

## 2. Material and Methods

### 2.1. Bacteria

The acute cystitis strain CY17 was isolated during a prospective study of childhood UTIs in Gothenburg, Sweden [12]. CY17 and other clinical isolates from this study are highly virulent in the murine acute cystitis model, as are other clinical isolates from the same patient group [2]. The prototype strain *E. coli* CFT073 (O6:K2:H1) [13] was used for the acute pyelonephritis infection model [3]. Four clinical ESBL-producing *E. coli* isolates (ESBL1, ESBL2, ESBL3, and ESBL4) were isolated from women with acute cystitis at the Department of Clinical Microbiology, Lund University Hospital, Sweden. The strains were phenotypically characterized for virulence factor expression (type-1 fimbriae, P fimbriae, and hemolysin production). ESBL1, ESBL2, and ESBL3 were type-1 fimbriae positives. ESBL4 was type-1 and P-fimbriae positive. All strains were hemolysin positive. Bacteria were cultured on trypsin soy agar (TSA) plates at 37 °C for 16 h, harvested in phosphate-buffered saline (PBS, pH 7.2), and appropriately diluted for infection.

### 2.2. Mice

Female C57BL/6J *WT* mice, C57BL/6J *Asc*^−/−^ or C57BL/6J *Irf3*^−/−^ mice were used at 9–15 weeks of age. *Asc*^−/−^ mice were originally provided by the laboratory of J. Tschopp, Department of Biochemistry, University of Lausanne, and the Institute for Arthritis Research (aIAR). *Irf3*^−/−^ mice were kindly provided by the Riken Bioresource Center, Japan, with permission from T. Taniguchi [14]. Mice were bred and housed in a specific pathogen-free animal facility (Lund University, Lund, Sweden). The mice had free access to food and water.

### 2.3. Study Approval

Experiments were approved by the Malmö/Lund Animal Experimental Ethics Committee at the Lund District Court, Sweden (#M119-16 and 6551-2021). Animal care and protocols followed institutional, national, and European Union guidelines and were governed by the European Parliament and Council Directive (2016/63, EU), the Swedish Animal Welfare Act (Djurskyddslagen 1988:534), the Swedish Welfare Ordinance (Djurskydssförordningen 1988:539), and the Institutional Animal Care and Use Committee (IACUC) Guidelines.

### 2.4. Experimental Acute Cystitis

Mice were anesthetized by intraperitoneal (i.p.) injection of a cocktail of ketamine (1.48 mg in 100 μL of 0.9% NaCl solution, Intervet) and xylazine (0.22 mg in 100 μL of 0.9% NaCl solution, Vetmedic) and infected by intravesical inoculation with *E. coli* CY17, *E. coli* ESBL1, *E. coli* ESBL2, *E. coli* ESBL3, *E. coli* ESBL4, or *E. coli* CFT073 through a soft polyethylene catheter (2 × 10^9^ CFU/mL, 50 μL) [2]. IL-1RA (anakinra, 1 mg/kg in saline) or cefotaxime (100 mg/kg in saline) was injected i.p. 6 h post-infection and daily for 7 days. Urine samples were obtained before infection and at regular intervals after infection (24 h, 3 days, 5 days, and 7 days). Bacteria were cultured on TSA plates overnight, and urine neutrophils were quantified in a hemocytometer chamber. At sacrifice, organs were aseptically removed and photographed for gross pathology analysis. Tissue samples were fixed in 4% PFA and embedded in paraffin for hematoxylin and eosin staining (H&E) and immunohistochemistry (IHC).

Pain behavior (locomotion and rearing) was measured for each mouse on day 7, using a clear cage and a three-minute recording device, as described by Ruddick et al. [15]. For long-term experiments, *Asc*^−/−^ were infected with *E. coli* ESBL4 and treated with IL-1RA (1 mg/kg) or cefotaxime (100 mg/kg) daily for seven days. Urine samples were obtained at regular intervals before sacrifice at day 14 or day 28 (day 1, day 3, day 5, day 7, day 10, day 14, day 21, and day 28).

### 2.5. Bladder Pathology Scoring

Gross bladder pathology was visually assessed at sacrifice. Scoring was based on the degree of hyperemia and edema on a scale from 0–10, where 0 was unchanged compared to uninfected controls and 10 was the most extensive change recorded. Scoring was performed independently by two researchers but could not be fully blinded due to the large differences between the groups.

### 2.6. Immunohistochemistry

Whole bladder paraffin sections (4 μm thick) were mounted on positively charged microscope slides, de-paraffinized in xylene and ethanol washes, and subjected to heat-induced antigen retrieval. Samples were washed (PBS), permeabilized (0.25% Triton X-100 in PBS), and blocked (1% bovine serum albumin [BSA] in 5% fetal calf serum [FCS] in PBS), incubated with primary antibody at 4 °C overnight, washed (0.025% Triton X-100 in PBS), and stained with appropriately labeled secondary antibodies. The nuclei were counterstained with DAPI, washed, and mounted using Fluoromount. Sections were imaged by laser scanning confocal microscopy (Zeiss LSM 800, Zeiss, Jena, Germany). Polyclonal rabbit anti-NK1R (1:50, sc-15323, Santa Cruz Biotechnology, Santa Cruz, CA, USA) or monoclonal rat anti-substance P (1:50, sc-21715, Santa Cruz Biotechnology) antibodies were used.

### 2.7. Western Blotting

Proteins from whole bladder tissue were extracted using NP40 lysis buffer with added protease and phosphatase inhibitors. Extracted proteins (10 µg) were loaded and run on NuPage gels, blotted onto PVDF membranes, blocked (10% non-fat dry milk), stained using polyclonal rabbit anti-NK1R (1:200, Santa Cruz Biotechnology), or mouse monoclonal anti-β-actin (1:4000, Thermo Fisher Scientific, Waltham, MA, USA) in 10% BSA. Blots were washed in PBS-T (PBS-0.1% Tween 20, 3 × 10 min), stained with HRP-labeled goat anti-rabbit (1:4000, BioRad, Hercules, CA, USA) or HRP-labeled rabbit anti-mouse (1:4000, DAKO, Agilent Technologies, Glostrup, Denmark) in 10% milk, washed again (3 × 10 min), incubated with ECL reagent, and analyzed in the GelDoc imaging system (BioRad).

### 2.8. Urine Cytokine and Chemokine Measurements

Urine samples were analyzed using mouse IL-1β or mouse CXCL1 DuoSet kits. Substance P was analyzed by the Substance P Parameter Kit, as per the manufacturer’s instructions (all from R&D Systems, Minneapolis, MN, USA).

### 2.9. Global Gene Expression in Infected Bladders

Total RNA was extracted from murine bladders with the RNeasy Mini Kit (Qiagen, Hilden, Germany) after disruption in RLT buffer with added β-mercaptoethanol (1%) using the Precellys Lysing Kits (Bertin Technologies, Montigny-le-Bretonneux, France) and Tissuelyser (Qiagen).

Total bladder RNA was amplified using the GeneChip 3′ IVT Express Kit, hybridized onto Mouse Genome 430 PM array strips (16 h at 45 °C), washed, stained, and scanned using the GeneAtlas system (Affymetrix, Santa Clara, CA, USA). The data were normalized using the robust multi-average implemented in the Transcriptomics Analysis Console software (Applied Biosystems, TAC v.4.0.1.36). Significantly altered genes were sorted by relative expression (2-way ANOVA model using Method of Moments, *p*-values < 0.05, and absolute fold change > 4) and analyzed using Ingenuity Pathway Analysis software (Ingenuity Systems, Qiagen Bioinformatics). Heat maps were constructed using the Gitools 2.1.1 software (the Biomedical Genomics Group, Barcelona, Spain).

### 2.10. Statistics

Data were analyzed for the Gaussian distribution defined by the D’Agostino and Pearson normality test; no data were considered normally distributed. Non-parametric data was analyzed by the Mann–Whitney U-test or by using the Kruskal–Wallis tests. *p* < 0.05 was considered significant. The data were examined using Prism (v. 6.0 GraphPad, Boston, MA, USA).

## 3. Results

### 3.1. IL-1RA Treatment of Acute Cystitis in Susceptible Asc^−/−^ Mice

Acute cystitis was established in susceptible *Asc*^−/−^ mice by infection with the clinical acute cystitis isolate *E. coli* CY17, and the mice were sacrificed on day 7 (schematic in Figure 1A). Infected mice developed severe acute cystitis, characterized by edema, hyperemia, and a loss of tissue structure (Figure 1B,C). Urine cultures remained positive until sacrifice with no evidence of bacterial clearance (Figure 1D,E). A dramatic increase in urine neutrophil numbers was detected after 24 h, and neutrophil counts remained high until sacrifice (Figure 1F,G). Urine IL-1β and CXCL1 levels were elevated compared to uninfected mice (Figure 1H).

Infected mice were treated with IL-1RA (1 mg/kg, i.p. injection, 6 h post-infection and daily for seven days) or with cefotaxime (100 mg/kg), using the same protocol (Figure 1A). PBS was used in the sham-treated group. IL-1RA treatment accelerated bacterial clearance to the same extent as cefotaxime (Figure 1D,E). Treated mice showed a marked reduction in gross pathology scores (*p* < 0.001 compared to infected controls, Figure 1B,C). Neutrophil infiltration was inhibited after 24 h and further reduced after seven days (Figure 1F,G). Inflammatory markers in urine (IL-1β and CXCL1) were reduced to background levels (Figure 1H). Interestingly, IL-1RA-treated mice had significantly lower gross pathology scores than cefotaxime-treated mice, lower neutrophil numbers, and lower IL-1β and CXCL1 levels in urine (Figure 1C,F–H), consistent with a combined anti-bacterial and anti-inflammatory effect of IL-1RA treatment. A direct anti-bacterial effect was excluded by culturing CY17 in vitro in the presence of IL-1RA (1 mg/mL). In contrast, cefotaxime (10 µg/mL) reduced bacterial numbers in vitro after 4 h (Appendix A).

### 3.2. IL-1RA Treatment of Acute Cystitis Caused by Antibiotic Resistant E. coli Strains

To determine if the therapeutic efficacy of IL-1RA extends to antibiotic-resistant strains, *Asc*^−/−^ mice were infected with ESBL-producing *E. coli* isolates using the protocol described above (Figure 2). Infection caused acute cystitis, according to the criteria described above. IL-1RA treatment was shown to reduce gross pathology scores and accelerate bacterial clearance from day 3 (Figure 2A–E). Urine neutrophil numbers were reduced by 50–70% after 24 h and remained low until sacrifice, confirming the therapeutic efficacy of IL-1RA treatment (Figure 2F,G). Urine IL-1β levels were strongly reduced (Figure 1H). Cefotaxime treatment had no effect on these endpoints compared to the sham-treated group (Figure 2A–H).

The infection with ESBL-producing *E. coli* was repeated in *Asc*^−/−^ mice using 3 additional clinical ESBL isolates (Figure 2I–K). Bacterial numbers were significantly reduced by the IL-1RA treatment. A pronounced anti-inflammatory effect of IL-1RA was quantified as the number of PMNs in urine. The infection with ESBL-producing *E. coli* was repeated in C57BL/6J WT mice, which are more resistant to bladder infection than *Asc*^−/−^ mice. The same four different ESBL strains were used for infection (Appendix A). C57BL/6J WT mice developed transient acute cystitis with lower gross pathology scores, neutrophil recruitment, and bacterial counts than *Asc*^−/−^ mice. A significant effect of IL-1RA treatment was still detected in the C57BL/6J WT mice (Appendix A).

### 3.3. Long-Term Effects of IL-1RA Therapy

To determine if IL-1RA therapy has a lasting effect, susceptible *Asc*^−/−^ mice were infected with *E. coli* CY17, treated for seven days with IL-1RA (1 mg/kg) or cefotaxime (100 mg/kg), and followed without further treatment until day 14, when all parameters were re-investigated (Figure 3A–D). There was no evidence of a relapse of infection in IL-1RA-treated mice. Gross pathology scores and tissue bacterial counts remained low. Cefotaxime treatment showed similar extended effects as IL-1RA, suggesting that CY17 infection was cleared (Figure 3A–D).

A marked difference in treatment outcome was detected in *Asc*^−/−^ mice infected with the ESBL4 *E. coli* isolate. At long-term follow-up on days 14 and 28, IL-1RA-treated mice remained protected (Figure 3E–G). There was no increase in bacterial counts or inflammatory parameters compared to day 7 (Figure 3H,I). Cefotaxime-treated mice, in contrast, showed gross bladder pathology with edema and hyperemia, comparable to infected *Asc*^−/−^ mice that had not received treatment. Urine neutrophil counts and bacterial counts remained positive, and bladder tissues remained infected (Figure 3E–I). The results suggest that, unlike cefotaxime, IL-1RA treatment has a lasting effect on ESBL-producing strains in mice with acute cystitis.

### 3.4. Inhibition of the Pain Response in Acute Cystitis

Acute cystitis is accompanied by pain due to direct bacterial nerve cell activation of the pain sensors substance P and NK1R [4]. The pain response is potentiated by an IL-1β-dependent activation loop of epithelial origin [4]. Tissue levels of substance P and NK1R were increased in infected *Asc*^−/−^ mice (Figure 4A), and NK1R showed significant co-localization with substance P in the basal epithelial layer, suggesting that the pain signal might be propagated from the mucosa to the central nervous system. 

The pain behavior was more strongly affected in IL-1RA-treated mice than in cefotaxime-treated mice, compared to the sham-treated group (Figure 4B,C). All treated mice showed a return to normal movement behavior, but effects on rearing indicated some remaining discomfort in cefotaxime-treated mice (*p =* 0.17 compared to infected control mice). NK1R and substance P staining were low, with little evidence of co-localization in treated mice (Figure 4A). Urine substance P levels were more strongly inhibited by IL-1RA than cefotaxime (*p* = 0.002, Figure 4D), and by Western blot analysis of whole bladder extracts, NK1R levels were lower in IL-1RA-treated than in cefotaxime-treated mice (Figure 4E).

### 3.5. Effects on Gene Expression in Infected Bladders

To further characterize the treatment effects, whole bladder mRNA was subjected to genome-wide transcriptomic analysis (Figure 5). Bladder RNA from CY17-infected *Asc*^−/−^ mice was first compared to uninfected controls (day 7, cut-off FC ≥ 2, *p* < 0.05). In untreated mice, a total of 1566 significantly regulated genes were identified, with a predominance of genes regulating neutrophil recruitment and activation (Figure 5A–D). IL-1β and inflammasome-related genes were strongly upregulated, including *Il1b* itself (FC = 50.6) and *Mmp7* (FC = 29.8), which encodes the matrix metalloproteinase (MMP)-7 protease responsible for IL-1β processing in *Asc*^−/−^ mice (Figure 5B,E) [2]. Furthermore, *Cxcl3* (FC = 41.4) and the antimicrobial proteins *S100a9*, *S100a8* (FC = 204.3 and 90.7), and *Ccl3l1* (FC = 28.5) were strongly upregulated by infection (Figure 5B). An IL-1- and MyD88-centric gene network was identified as strongly activated in CY17-infected mice compared to uninfected control mice (Figure 5C). *Asc* (*Pycard*) expression was upregulated in the CY17-infected, untreated mice but not in the treatment groups, confirming the regulation of pro-IL-1 processing by infection (Figure 5E).

IL-1RA or cefotaxime treatment markedly reduced gene expression in CY17-infected mice, including the top-regulated genes described above (Figure 5A,B). Biofunction analysis revealed significant differences in gene expression between the untreated controls and cefotaxime- or IL-1RA-treated mice (Figure 5D). Neutrophil migration, granulocyte adhesion, and phagocyte infiltration were strongly inhibited by IL-1RA treatment but not by cefotaxime treatment. IL-1RA treatment markedly reduced the expression of IL-1β and inflammasome-related genes, and IL-1- and MyD88-centric networks were no longer activated in the IL-1RA-treated mice (Figure 5E). Cefotaxime treatment reduced gene expression to a similar extent as IL-1RA but did not inhibit neutrophil migration and activation (Figure 5D). *Tnf* and *Il18* remained activated in the cefotaxime-treated group, unlike the IL-1RA-treated mice, as did *Nlrp3*, which is essential for canonical pro-IL-1β processing by the inflammasome (Figure 5E).

The ESBL4 strain was less virulent than CY17, and gene expression was less strongly activated in infected mice (Figure 6A). IL-1RA treatment inhibited gene expression to a significant extent, including IL-1 β, IL-1R1, and inflammasome-related genes (Figure 6B–D). In contrast, innate immunity was activated with cefotaxime treatment, including *Il6*, *Il33*, *Cxcl3,* and *Ptgs2*, consistent with the edema and hyperemia of the bladders from cefotaxime-treated mice (Figure 6B). Cefotaxime activated a higher pro-inflammatory response in ESBL-infected mice, suggesting effects on the production of pro-inflammatory bacterial metabolites and/or host response pathways (Figure 6C–E). 

### 3.6. IL-1RA Treatment Protects Irf3^−/−^ Mice from Acute Pyelonephritis

Since the IL-1 response was strongly regulated in infected bladders, we further investigated if IL-1RA might be protective in infected kidneys (schematic in Figure 7A). *Irf3*^−/−^ mice are highly susceptible to kidney infection and develop acute pyelonephritis, with high bacterial counts and excessive innate immune activation, leading to abscess formation [3]. *Irf3*^−/−^ mice infected with the uropathogenic *E. coli* strain CFT073 developed severe acute pyelonephritis with macroscopically visible gross pathology on day seven, defined by extensive abscess formation, hyperemia, and edema, elevated bacterial counts in renal tissues, and extensive neutrophil recruitment (Figure 7).

IL-1RA was administered intra-peritoneally to *Irf3*^−/−^ mice, directly post-infection and daily for seven days. A marked reduction in gross pathology was observed after seven days, defined by a lack of abscess formation, hyperemia, and edema in treated *Irf3*^−/−^ mice compared to the sham-treated group (Figure 7B). Bacterial counts were markedly reduced in infected kidneys, bladders, and urine (Figure 7C–E). Neutrophil recruitment was inhibited (Figure 7F,G) compared to untreated *Irf3*^−/−^ mice, suggesting accelerated bacterial clearance.

### 3.7. Effects of IL-1RA Treatment on Kidney Infections Caused by Antibiotic-Resistant E. coli Strain

*Irf3*^−/−^ mice were subsequently infected with the ESBL4 *E. coli* strain using the same protocol (Figure 8A). Cefotaxime treatment was used as a comparator, and PBS was used in the sham-treated group. The disease phenotype in ESBL-infected *Irf3*^−/−^ mice was less severe than in mice infected with the uropathogenic *E. coli* strain CFT073, for all disease-related parameters, reflecting the difference in virulence between those strains (Figure 8B–D). 

*Irf3*^−/−^ mice infected with the ESBL4 strain were further treated with IL1-RA using the protocol described above. A significant protective effect was detected after three days compared to the untreated group and remained highly significant after seven and 14 days (Figure 8D–H). Bacterial counts were markedly reduced in infected kidneys, bladders, and urine (Figure 8E,G–H), and neutrophil recruitment was inhibited (Figure 8F), compared to untreated *Irf3*^−/−^ mice, suggesting accelerated bacterial clearance and inhibition of inflammation. A corresponding difference in bacterial counts and neutrophil recruitment was detected in infected bladders. Importantly, the *Irf3*^−/−^ mice do not develop bladder pathology comparable to the *Asc*^−/−^ mice (Appendix A).

## 4. Discussion

Immunotherapy offers a new approach to treating bacterial infections. This study identifies IL-1RA therapy as a promising treatment option for acute cystitis and acute pyelonephritis, caused by *E. coli* infections. Rather than killing bacteria directly, IL-1RA therapy boosted the antibacterial host defense and inhibited the excessive inflammatory response that accompanies both infections and causes tissue pathology. The findings illustrate how correcting immune imbalances may restore the antibacterial defense in susceptible hosts and the ability to limit the destructive effects of pathogen attack and clear the infection. Our findings further provide proof-of-concept in mice that immunomodulation may have a therapeutic effect on multidrug-resistant bacterial strains, which is essential. Furthermore, the IL-1 receptor antagonist was shown to efficiently reduce inflammation, pathology, and pain.

Cefotaxime is a classical antibiotic and member of the cephalosporin family and is bactericidal due to its effects on bacterial cell wall synthesis [16]. This was clear from in vitro experiments in this study, where bacterial growth was inhibited and bacterial numbers were reduced in broth supplemented with cefotaxime. Interestingly, no direct bactericidal effects were seen for IL-1RA. In vitro studies did not detect any bactericidal effect in liquid medium supplemented with IL-1RA at the concentration equivalent to the one used to treat the mice, suggesting that the host is responsible for the antibacterial effect in IL-1RA-treated mice. These findings may appear counterintuitive, as immunomodulation would be expected to impair rather than augment the resistance to bacterial infection, but the effects of IL-1RA treatment were highly significant and clear-cut. Importantly, in this case, immunomodulation did not create an immunodeficiency state but restored immune balance in hosts, where disease was caused by excessive immune activation. Interfering with immune dysregulation may thus offer a more specific solution to limiting inflammation and pain compared to general anti-inflammatory agents, including non-steroidal anti-inflammatory drugs and COX-2 inhibitors, as well as corticosteroids. The present study suggests that by targeting dysregulated molecular circuits, immunomodulation may have a more specific effect than general anti-inflammatory agents.

Immunomodulatory drugs are commonly used to balance overactive immune responses in patients with auto-inflammatory and autoimmune disorders. IL-1 engages the receptor IL-1R1 and forms a heterodimer with the IL-1 receptor accessory protein (IL-1RAcP), activating downstream signaling, including pro-inflammatory cytokines, chemokines, and pain mediators [10]. The recombinant human IL-1RA inhibits hetero-dimerization and downstream signaling, attenuating IL-1 hyper-activation, inflammation, and pain in the murine acute cystitis model [2,4]. Recent observations have provided strong evidence that IL-1RA treatment may be a useful therapeutic option in cystic fibrosis, and IL-1RA-treated patients suffering from severe SARS-CoV-2 infection show evidence of reduced inflammation and decreased mortality compared to patients receiving standard care [17,18]. The efficiency of IL-1RA therapy in bacterial infections was first observed in *Asc*^−/−^ mice, which are susceptible to acute cystitis due to an IL-1 hyper-activation disorder. The present study extends the therapeutic window to include acute pyelonephritis, in this case in *Irf3*^−/−^ mice, which are susceptible to infection due to a transcription factor deficiency that impairs the type I IFN response and the antibacterial defense. The potent effects of IL-1RA suggest that the IL-1 cytokine family is a major contributor to pathology both in acute cystitis and during renal infection.

Pain is a hallmark of acute cystitis. The activation of SP, NK1R, and other pain mediators generates a peripheral pain signal and propagates it from local afferent nerves at the site of infection to the dorsal roots of the central nervous system [4,19]. The contribution of IL-1 to pain was confirmed here, as IL-1RA inhibited the activation of SP and NK1R in nerve cells and the pain-related behavior in infected mice. Clinical data from patients with bladder pain syndrome further support this treatment concept [20]. The patients suffer from severely handicapped chronic pain from the urinary bladder and have elevated urine SP levels, suggesting that chronic nerve cell activation may maintain the sensation of severe pain [21,22,23]. In a recent clinical study, IL-1RA treatment was associated with symptom relief and an improved quality of life in about 50% of treated patients. Urine SP levels were lowered, and IL-1-dependent gene networks were inhibited, confirming the effects on the host response [20]. A placebo-controlled phase II study has been initiated in this patient group.

UTIs are among the most prevalent infectious diseases globally [24,25], most often caused by uropathogenic *Escherichia coli* (UPEC) originating from the gut- or perineal flora. Resistance is spreading rapidly [1,26,27,28]. Antibiotic-resistant *E. coli* infections account for one-half of the estimated global burden of antibiotic resistance, with about 90% of *E. coli* strains being resistant to at least one antibiotic [29,30,31]. In Europe, UPEC isolates are resistant to third-generation cephalosporins and fluoroquinolones [32] (11.8% and 22.3%, respectively), and fluoroquinolone-resistant UPEC represented 31.3% of the isolates in hospitalized UTI patients [33,34,35,36]. Unexpectedly, IL-1RA treatment accelerated bacterial clearance, with similar antibacterial efficacy as cefotaxime with the added benefit of also removing antibiotic-resistant *E. coli* strains. The effects on acute disease were highly significant, and long-term follow-up suggested that IL-1RA therapy might have a lasting protective effect. The findings are conceptually important, as the bacteria were killed at least as efficiently by immunotherapy as by antibiotics, with the added benefit of a broader anti-bacterial spectrum and reduced inflammation.

## 5. Patents

Patents, with the scientists as inventors, have been filed for the use of IL-1RA as a treatment for cystitis and bladder pain.

## Figures and Tables

**Figure 1 pathogens-13-00042-f001:**
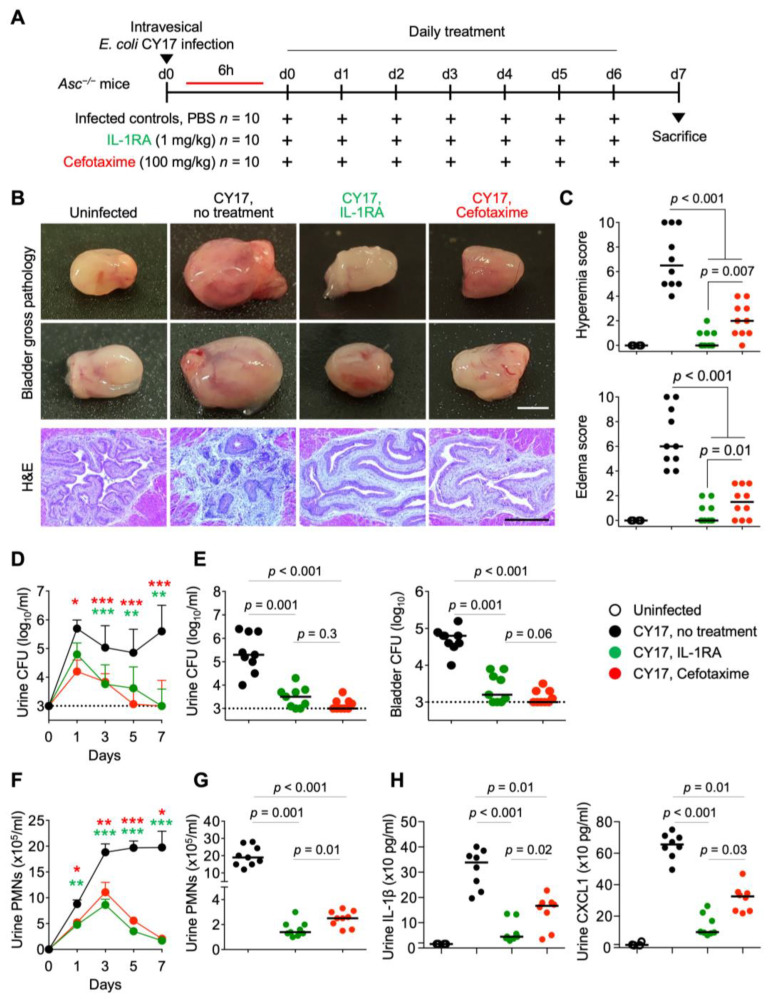
Efficacy of IL-1RA treatment in acute cystitis compared to cefotaxime. (**A**) Schematic of the infection and treatment protocol. Genetically predisposed *Asc*^−/−^ mice were infected with the acute cystitis isolate *E. coli* CY17 (antibiotic sensitive) at 10^8^ CFU. Six hours post-infection, mice were treated i.p. with IL-1RA (anakinra, 1 mg/kg) or cefotaxime (100 mg/kg). Treatment was continued daily, until day 7, when mice were sacrificed (*n =* 9 mice per group from 2 experiments). (**B**) Treatment alleviated severe cystitis in CY17-infected mice. Differences in gross bladder pathology between treated (IL-1RA or cefotaxime) and untreated *Asc*^−/−^ mice. (**C**) Gross pathology quantified by edema and hyperemia scores was reduced in treated mice compared to untreated mice. (**D**–**G**) Treatment accelerated bacterial clearance and reduced inflammation, quantified as neutrophil recruitment. (**D**) Kinetics of infection. Mean log_10_ CFU/mL urine is shown for days 1, 3, 5, and 7, post-infection. (**E**) Bacterial counts in urine and bladder tissue (day 7). (**F**) Kinetics of neutrophil recruitment. Mean neutrophil numbers in urine are shown for days 1, 3, 5, and 7, post-infection. (**G**) Quantification of neutrophils in urine (day 7). (**H**) Effects of treatment on inflammatory mediators IL-1β and CXCL1 in urine at day 7. Representative bladder images (**B**), scale bar = 2 mm. Data are presented as individual mice, and the horizontal lines represent the median (**C**,**E**,**G**,**H**) and mean ± SEM (**D**,**F**). Data were analyzed using the Kruskal–Wallis test, * *= p* < 0.05; ** = *p* < 0.01; *** = *p* < 0.001.

**Figure 2 pathogens-13-00042-f002:**
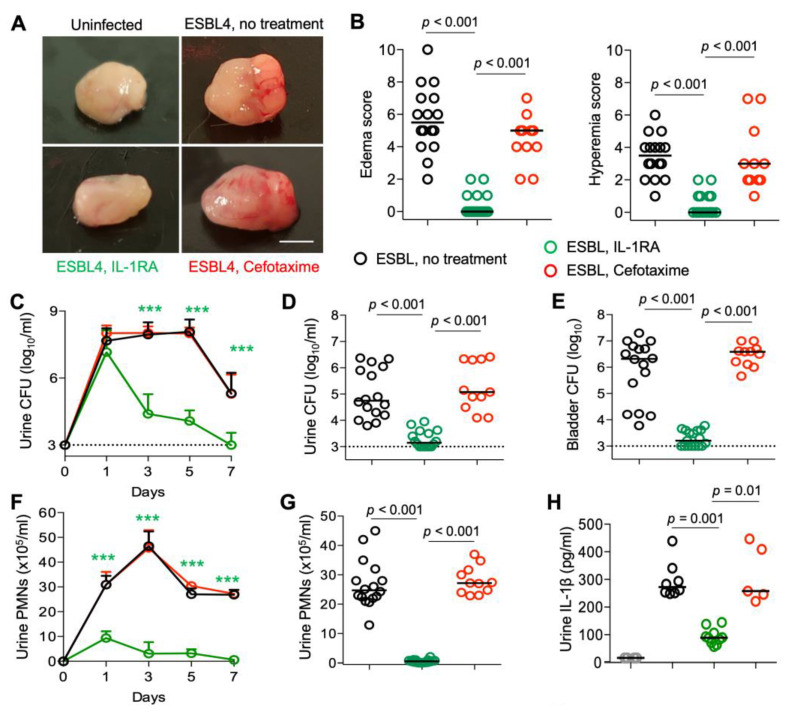
Efficacy of IL-1RA treatment in *Asc*^−/−^ mice infected with ESBL strains. *Asc*^−/−^ mice were infected with each of four ESBL-producing *E. coli* isolates (10^8^ CFU, *n* = 16 mice from 3 experiments) and treated according to the protocol shown in Figure 1A with IL1-RA (anakinra, 1 mg/kg, *n* = 17 mice) or cefotaxime (100 mg/kg *n =* 11 mice) daily for seven days. Data in A-H represent ESBL strain 4, data in I-K represent ESBL strains 1–3. (**A**,**B**) Therapeutic efficacy of IL-1RA defined by a loss of gross pathology, hyperemia, and edema. No treatment effect was seen in cefotaxime-treated mice. (**C**–**E**) Accelerated bacterial clearance in IL-1RA- but not cefotaxime-treated mice. (**C**) Kinetics of infection. Mean log_10_ CFU/mL urine is shown for days 1, 3, 5, and 7, post-infection. (**D**) Quantification of bacteria in urine (day 7). (**E**) Quantification of bacteria in whole bladder tissues (day 7). (**F**,**G**) Neutrophil recruitment inhibited by IL-1RA but not cefotaxime treatment. (**F**) Kinetics of neutrophil recruitment. Mean neutrophil numbers in urine are shown for days 1, 3, 5, and 7, post-infection. (**G**) Quantification of neutrophils in urine (day 7). (**H**) Inhibition of urine IL-1β responses by IL-1RA treatment but not by cefotaxime. (**I**–**K**) Kinetics of infection and neutrophil recruitment. Bacterial counts and neutrophil numbers in the urine of mice infected with ESBL strains 1–3. Data are presented as individual mice, and the horizontal lines represent the median (**B**,**D**,**E**,**G**,**H**) and mean ± SEM (**C**,**F**,**I**–**K**). Data were analyzed using the Kruskal–Wallis test, * = *p* < 0.05; ** = *p* < 0.01; *** = *p* < 0.001.

**Figure 3 pathogens-13-00042-f003:**
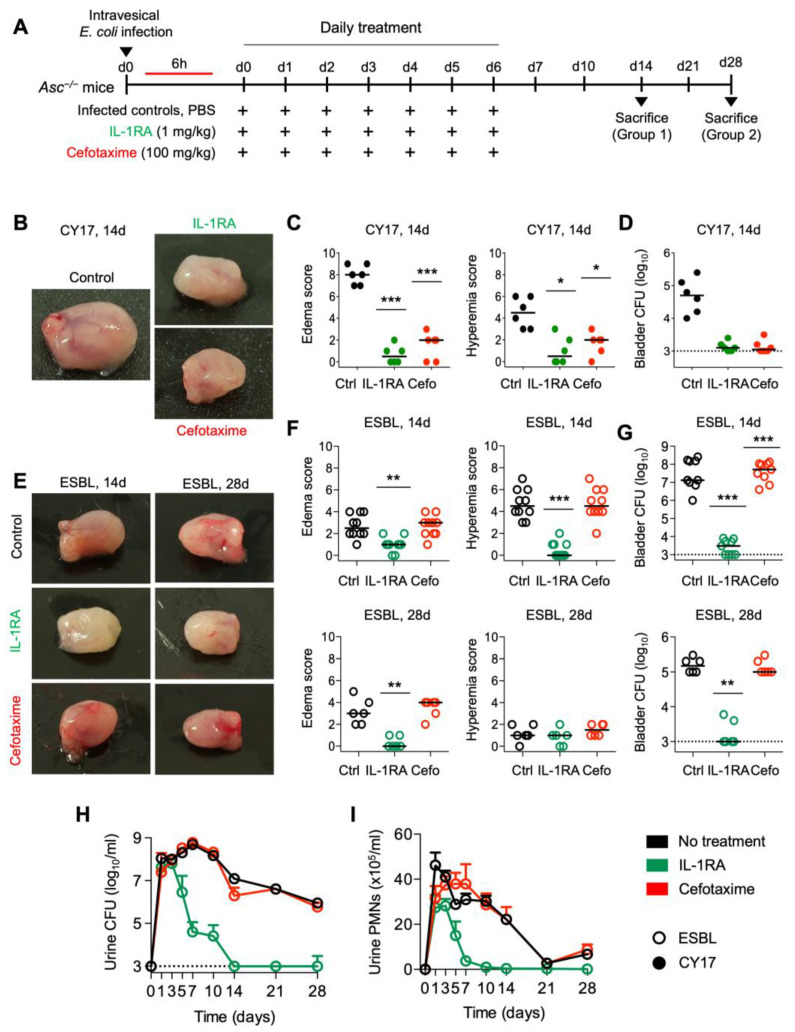
Long-term protection by IL-1RA treatment in *Asc*^−/−^ mice infected with CY17 and ESBL strains. (**A**) Schematic of the infection and treatment protocol. *Asc*^−/−^ mice were infected with the CY17 or the ESBL4 strain at 10^8^ CFU. Six hours post-infection, mice were treated i.p. with IL-1RA (anakinra, 1 mg/kg) or cefotaxime (100 mg/kg). Treatment was continued daily for 7 days, and the mice were followed without further intervention and sacrificed on day 14 or day 28 post-infection (*n* = 10 mice per group from 2 experiments). (**B**,**C**) IL-1RA treatment protected mice infected with CY17 from acute cystitis defined by low gross pathology scores and a lack of edema and hyperemia in treated compared to untreated mice. Cefotaxime treatment had similar treatment effect. (**D**) Bacterial counts for infection with the CY17 strain remained low in the treated groups. (**E**,**F**) IL-1RA treatment protected mice infected with the ESBL strain from acute cystitis, defined by low gross pathology scores and a lack of edema and hyperemia in treated compared to untreated mice. Cefotaxime treatment had no significant effect. (**G**) Log_10_ bacterial counts in bladder tissue 14- and 28-days post-infection. Bacterial counts for the ESBL strain remained low in the IL-1RA-treated groups. Cefotaxime treatment had no significant effect on the ESBL strain. (**H**) Kinetics of infection. Mean log_10_ CFU/mL urine is shown for days 1, 3, 5, 7, 10, 14, 21, and 28 post-infection. IL-1RA treatment accelerated bacterial clearance. Cefotaxime had no significant effect. (**I**) Kinetics of neutrophil recruitment. Mean neutrophil numbers in urine are shown for days 1, 3, 5, 7, 10, 14, 21, and 28 post-infection. IL-1RA treatment reduced inflammation, quantified as neutrophil recruitment. Cefotaxime did not significantly reduce neutrophil numbers. Representative bladder images (**B**,**E**), scale bar = 2 mm. Data are presented as individual mice, and the horizontal line represents the median (**C**,**D**,**F**,**G**) and mean ± SEM (**H**,**I**). Data were analyzed using the Kruskal–Wallis test, * = *p* < 0.05; ** = *p* < 0.01; *** = *p* < 0.001.

**Figure 4 pathogens-13-00042-f004:**
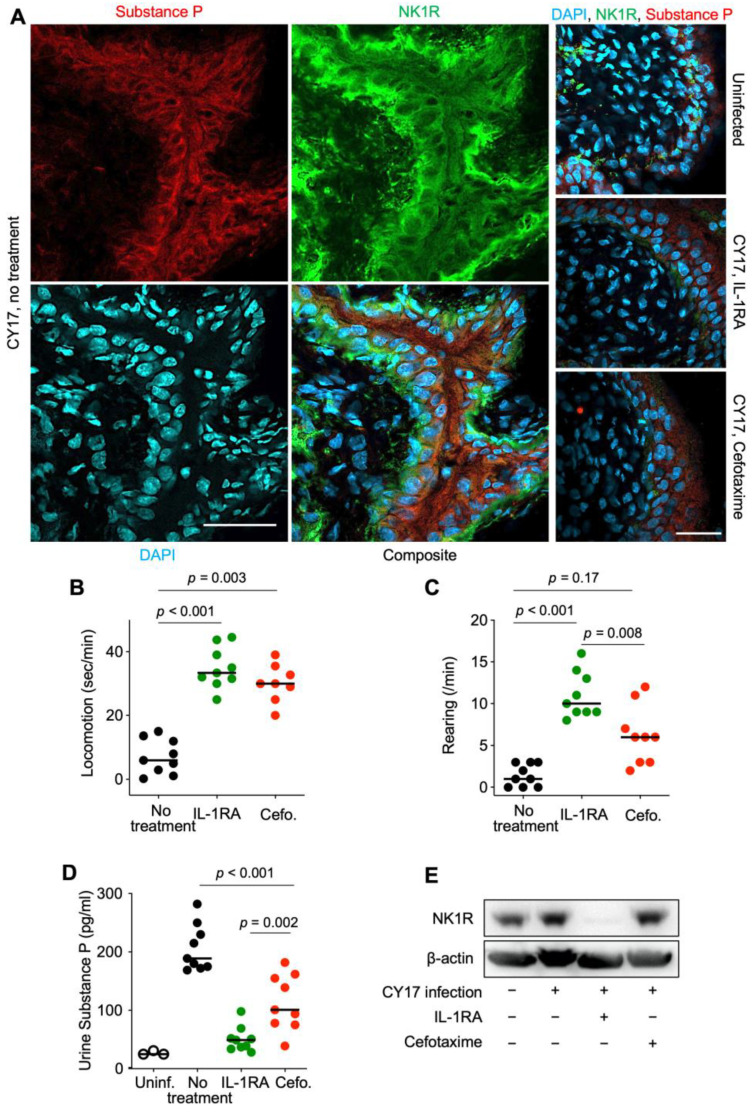
Effects of IL-1RA treatment on pain sensing in nerve cells and pain behavior in infected mice. (**A**) Effects of treatment on the pain sensors substance P (red) and NK1R (green). Bladder tissue sections (day 7) were processed for immunohistochemistry and nuclei were counterstained using DAPI (blue). (**B**,**C**) Pain behavior was recorded on day 7 in CY17-infected mice and was quantified as (**B**) the time of movement (seconds/min) and (**C**) the amount of rearing behavior (events/min). (**D**) Urine substance P levels in uninfected mice, infected untreated *Asc*^−/−^ mice, IL-1RA-treated, or cefotaxime-treated mice. (**E**) Western blot analysis of NK1R levels in whole bladder protein extracts (pooled samples from *n =* 4 mice per group). Representative images (**A**), *n* = 4 mice per group, scale bar = 2 mm. Data are presented as individual mice, and the horizontal line represents the median (**B**–**D**). Data were analyzed using the Kruskal–Wallis test.

**Figure 5 pathogens-13-00042-f005:**
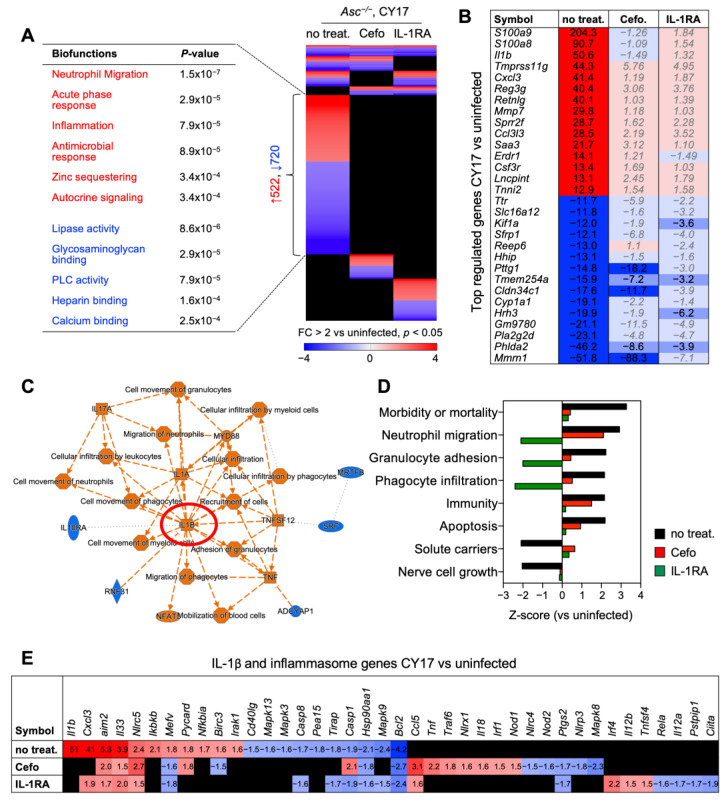
IL-1RA treatment inhibits the expression of pro-inflammatory genes in infected *Asc*^−/−^ mice. Whole bladder mRNA was subjected to transcriptomic analysis. (**A**) Gene expression profiles were compared between CY17-infected, untreated controls, and IL-1RA- or cefotaxime-treated mice (*n =* 2 mice per treatment group, *n* = 1 for uninfected control). Heat map of regulated genes in the treatment groups and CY17-infected controls compared to uninfected controls (cut-off FC ≥ 2, *p* < 0.05). Biofunctions of genes specific to CY17-infected controls included neutrophil migration, acute phase responses, inflammation, and antimicrobial responses. These responses were not activated in cefotaxime- or IL-1RA-treated mice (red = activated, blue = inhibited, black = not significantly regulated, compared to uninfected controls). (**B**) Top-regulated genes in CY17-infected mice treated with cefotaxime or IL-1RA compared to untreated controls. (**C**) IL-1 and MyD88-centric gene network in CY17-infected mice compared to uninfected controls (orange = activated function, blue = inhibited function). (**D**) Top-regulated biofunctions defined by Ingenuity Pathway Analysis (IPA) in CY17-infected untreated mice, compared to cefotaxime- or IL-1RA-treated mice (cut-off Z-score > ±1.5, *p*-value < 0.05). (**E**) Top-regulated IL-1β and inflammasome-regulated genes in CY17-infected mice. Activation of IL-1β and inflammasome-related genes in CY17-infected untreated mice and inhibition in IL-1RA- and cefotaxime-treated mice.

**Figure 6 pathogens-13-00042-f006:**
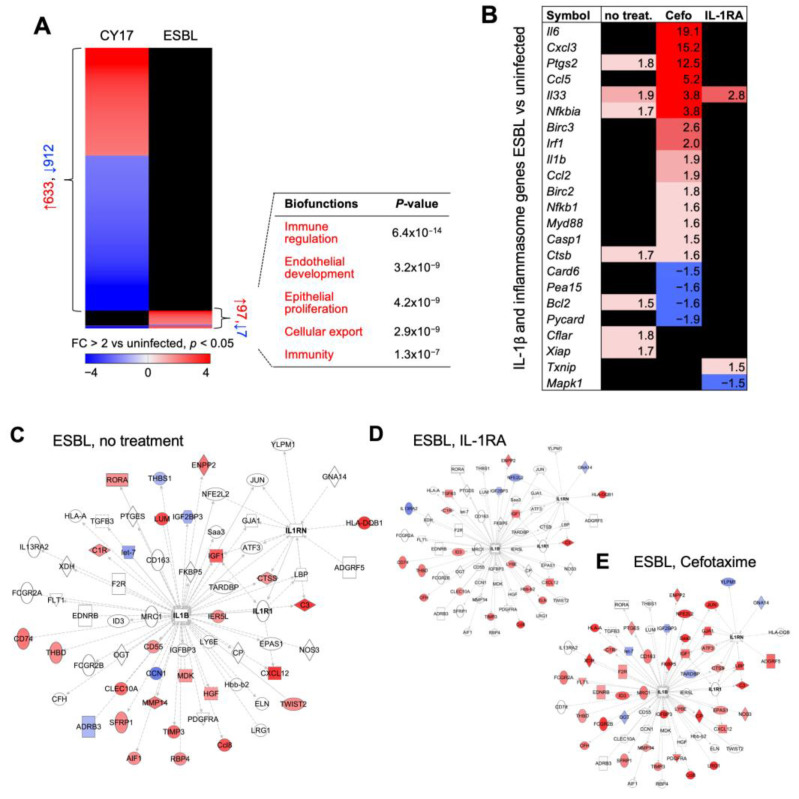
IL-1RA treatment inhibits and cefotaxime treatment activates the expression of pro-inflammatory genes in infected *Asc*^−/−^ mice. (**A**) Gene expression profiles were compared between CY17-infected and ESBL4-infected mice and uninfected controls (*n =* 2 mice per treatment group, *n* = 1 for uninfected control, cut-off FC ≥ 2, *p <* 0.05). A subset of ESBL-specific genes was identified and biofunctions were related to immune regulation, endothelial development, epithelial proliferation, and cellular export. (**B**) Heat map of IL-1β and inflammasome-regulated genes in the ESBL-infected treatment groups compared to uninfected controls (*n =* 2 mice per treatment group, *n* = 1 for uninfected control). (**C**–**E**) Limited effects of ESBL on gene expression, shown by an IL-1β and IL-RA-specific gene network (cut-off FC ≥ 2, *p <* 0.05). Gene networks from IL-RA or cefotaxime-treated mice are shown.

**Figure 7 pathogens-13-00042-f007:**
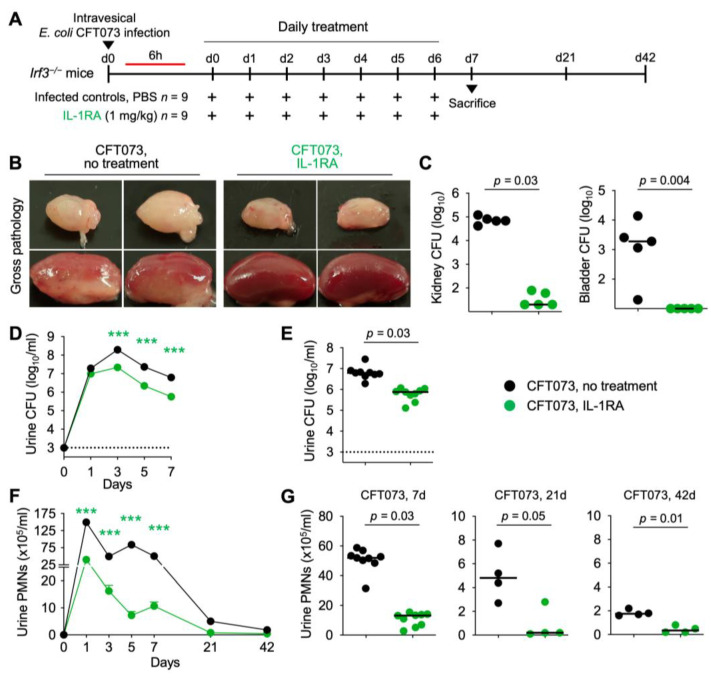
Efficacy of IL-1RA treatment in *Irf3*^−/−^ mice with acute pyelonephritis. (**A**) Schematic of the infection and treatment protocol. Genetically predisposed *Irf3*^−/−^ mice were infected with the uropathogenic strain *E. coli* CFT073 (antibiotic sensitive) at 10^8^ CFU. Six hours post-infection, mice were treated I.P. with IL-1RA (anakinra, 1 mg/kg). Treatment was continued daily, until day 7, when mice were sacrificed. Another group was followed for 21- and 42-days post-infection (*n* = 9 mice per group). (**B**) Therapeutic efficacy of IL-1RA defined by a loss of gross pathology, hyperemia, and edema. Treatment alleviated acute pyelonephritis in infected mouse kidneys. (**C**–**G**) Treatment accelerated bacterial clearance and reduced inflammation, quantified as neutrophil recruitment. (**C**) Bacterial counts in kidney and bladder tissue (day 7). (**D**) Kinetics of infection. Mean log_10_ CFU/mL urine is shown for days 1, 3, 5, and 7 post-infection. (**E**) Bacterial counts in urine (day 7). (**F**) Kinetics of neutrophil recruitment. Mean neutrophil numbers in urine are shown for days 1, 3, 5, 7, 21, and 42 post-infection. (**G**) Quantification of neutrophils in urine (days 7, 21, and 42). Representative bladder images (**B**). Data are presented as individual mice and the horizontal lines represent the median (**C**,**E**,**G**) and mean ± SEM (**D**,**F**). Data were analyzed using the Kruskal–Wallis test, *** = *p* < 0.001.

**Figure 8 pathogens-13-00042-f008:**
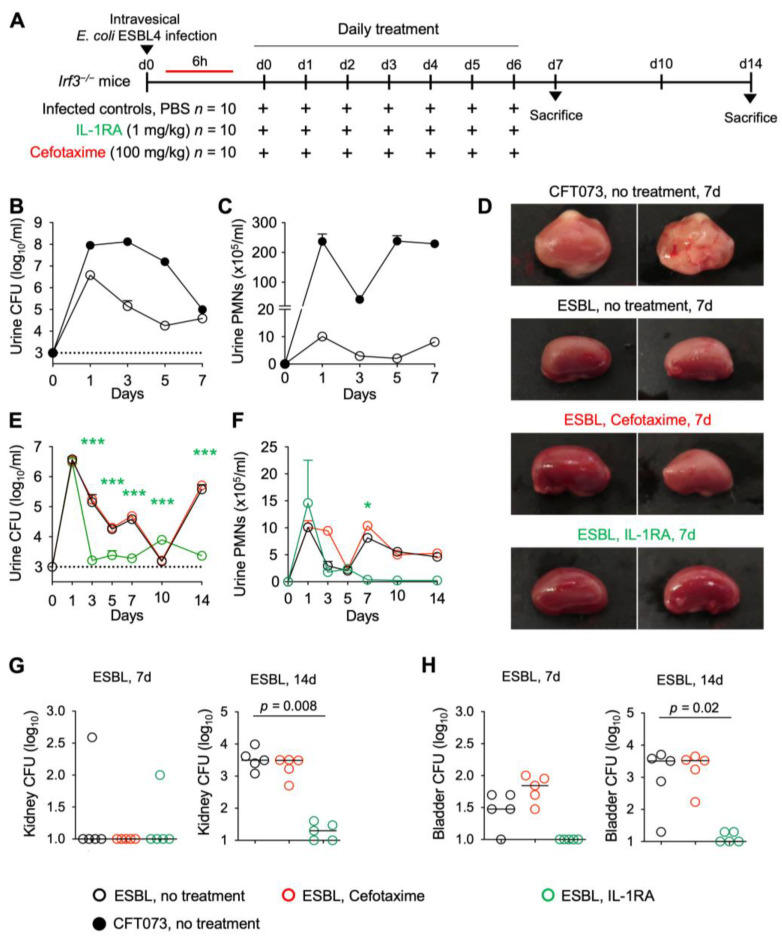
Efficacy of IL-1RA treatment on kidney infection in *Irf3*^−/−^ mice infected with an ESBL isolate. *Irf3*^−/−^ mice were infected with the ESBL4 *E. coli* isolate (10^8^ CFU, *n* = 10 mice from 1 experiment, 5 mice per time point) and treated according to the protocol shown in Figure 1A with IL-1RA (anakinra, 1 mg/kg, *n* = 10 mice) or cefotaxime (100 mg/kg *n* = 10 mice) daily for seven days. PBS was used in the control mice. (**A**) Schematic of the treatment protocol. (**B**,**C**) Comparison of mice infected with the uropathogenic strains CFT073 and ESBL4. (**D**) Therapeutic efficacy of IL-1RA defined by a loss of gross kidney pathology, hyperemia, and edema. No treatment effect was seen in cefotaxime-treated mice. (**E**) Kinetics of infection. Accelerated bacterial clearance in IL-1RA- but not cefotaxime-treated mice. Mean log_10_ CFU/mL urine is shown for days 1, 3, 5, 7, 10, and 14 post-infection. (**F**) Kinetics of neutrophil recruitment. Mean neutrophil numbers in urine are shown for days 1, 3, 5, 7, 10, and 14 post-infection. Neutrophil recruitment was inhibited by IL-1RA but not cefotaxime treatment. (**G**) Quantification of bacteria in whole kidney tissue (days 7 and 14). (**H**) Quantification of bacteria in whole bladder tissue (days 7 and 14). Data are presented as individual mice, and the horizontal lines represent the median (**G**,**H)** and mean ± SEM (**B**,**C**,**E**,**F**). Data were analyzed using the Kruskal–Wallis test, * = *p* < 0.05, *** = *p* < 0.001.

## Data Availability

The data supporting the findings of this study are available within the article and its Appendix A.

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
