# Peer review of "Therapeutic Effects of IL-1RA against Acute Bacterial Infections, including Antibiotic-Resistant Strains"

_pathogens, 2023, doi:10.3390/pathogens13010042_

Round 1

Reviewer 1 Report

Comments and Suggestions for Authors

The re-sults suggest that by targeting the IL-1 response, the control of the innate immune response to infection may be regained, with highly favorable treatment outcomes, including infections caused by antibiotic-resistant strains.

This is an elegant study with an important contribution to the field.
The impact of this work may be of much bigger amplitude, as the proposed
strategy could contribute across the board of infectious and non-infectious
pathologies.

Reviewer 2 Report

Comments and Suggestions for Authors

Comments to the authors:

Summary: The paper evaluates therapeutic effects of IL-1RA against acute bacterial infections, including antibiotic resistant strains. The authors suggest that by targeting the IL-1 response, the control of the innate immune response to infection may be regained, with highly favorable treatment outcomes, including infections caused by antibiotic-resistant strains. The overall content of the manuscript is well written. However, a few details in the introduction and discussion would be appreciated. I recommend the publication of this article after consideration of the comments below:

Introduction:

1.    IRF3/IRF7: Spell out the acronyms when used for first time in the document.

2.    Can authors elaborate on the functionality of the IRFs in the introduction in a bit more detail.

3.    Asc, Nlrp3, NKIR: Spell out the acronyms when used for first time in the document.

4.    Maintain same font across the manuscript. Please check Material and Methods under Bacteria section.

5.    Alignment if Figure 3 can be improved.

6.    Can authors elaborate on the mechanism of action of cefotaxime and how IL-1RA treatment is distinct and beneficial.

7.    Can authors please comment on how many antibiotic resistant bacterial strains were tested for IL1-RA treatment.

8.    IL1-RA treatment for antibiotic resistant bacterial urinary tract infections is exciting. Can authors comment on other bacterial infections?

9.    Authors mentioned that “Genetic screens have identified single genes that determine the susceptibility to acute pyelonephritis or acute cystitis”. Can authors comment on any down regulation of these genes in the case of IL1-RA treatment.

1. Can authors elaborate on differential gene expression with IL1-RA vs cefotaxime treatments for antibiotic susceptible and resistant bacterial strains.
